# Pumpkin (*Cucurbita* spp.): A Crop to Mitigate Food and Nutritional Challenges

**Monir Hosen** [1,2], **Mohd Y. Rafii** [3,*], **Norida Mazlan** [4], **Mashitah Jusoh** [3], **Yusuff Oladosu** [1], **Mst. Farhana Nazneen Chowdhury** [3,5], **Ismaila Muhammad** [1] and **Md Mahmudul Hasan Khan** [1,6]

1   Institute of Tropical Agriculture and Food Security, Universiti Putra Malaysia, Serdang 43400, Malaysia; monirhosendae81@gmail.com (M.H.); oladosuy@gmail.com (Y.O.); ismuha2000@gmail.com (I.M.); mhasan.bari12@gmail.com (M.M.H.K.)
2   Administration and Finance Wing, Department of Agricultural Extension, Khamarbari, Dhaka 1215, Bangladesh
3   Department of Crop Science, Faculty of Agriculture, Universiti Putra Malaysia, Serdang 43400, Malaysia; mashitahj@upm.edu.my (M.J.); farhananazneen80@gmail.com (M.F.N.C.)
4   Department of Plant Protection, Faculty of Agriculture, Universiti Putra Malaysia, Serdang 43400, Malaysia; noridamz@upm.edu.my
5   Department of Biochemistry, Sher-e-Bangla Agricultural University, Dhaka 1207, Bangladesh
6   Bangladesh Agricultural Research Institute (BARI), Gazipur 1701, Bangladesh
*   Correspondence: mrafii@upm.edu.my; Tel.: +60-3976-91043

**Abstract:** The world's food and agricultural programs have gradually declined into an unsustainable situation due to challenges such as increase in world population, varied agro-climatic regions, increase temperature, extreme sole-culture growing techniques, and water shortage. A considerable emphasis has been put on few staple food crops coupled with repeated dieting, food scarcity, and essential mineral deficits, frequently inducing dietary disorders. Because relying on staple crops may lead to serious food shortages in the future, we must adjust our dietary habits to include a diverse range of non-staple foods and maximize their use in order to achieve food security and reduce the nutritional gap. To assure healthy meals around the world, an authentic and reasonable strategy is presented to draw additional awareness towards variations in agricultural production techniques and dietary preferences. The EAT-Lancet declaration highlighted the importance of increasing agri-based foods to achieve sustainable health. Expanding overlooked crops with abundant genetic stocks and possibly beneficial characteristics is an approach that might meet food and nutritional security challenges. Although undervalued, pumpkin is a valuable vegetable herbaceous plant that contributes to global food and nutritional security. This crop has already been identified as a revolutionary age crop, balanced food, and more adapted to low soil and atmospheric circumstances than other major crops. This review paper focuses on the potential uses of pumpkin as an underutilized crop; diversification and development of hybrids, particularly hybridization breeding through diallel mating design; and how implementation of this "modern" technology would contribute to the breeding of the neglected pumpkin vegetable and stimulate productivity and nourish the world's largest malnourished, deprived, and starved populations.

**Keywords:** *Cucurbita* spp.; genetic diversity; breeding; hybridization; diallel mating





## 1. Introduction

In an expanding global and interconnected world, eliminating poverty and starvation is a moral necessity and a necessary precursor for international peace and security. The challenge of feeding the anticipated 9 billion people sustainably by 2050 could be encountered, among other things, by trying to rescue and using more variability in food and agriculture production systems, both in contexts of crops and cultivars within each crop [1,2]. The world's population presently gets most of its calories from a small number of crops, with only roughly 30 species accounting for 95% of world food energy [3]. On

the other hand, about 7000 species are recognized as palatable and some are even partially or fully domesticated, implying that a significant portion of potential food sources is underutilized [4,5]. Contemporary agricultural advancements, particularly prior to the Green Revolution, amplified this marginalization by focusing on a set of staple foods like wheat, maize, and rice, on which a large portion of the global population had already relied for food and nutritional security, as well as the areas where priority investments were made [6]. For this and many other factors, a number of consumable species of plants are considered insignificant, underutilized, or ignored, and have become part of the NUS (neglected and underutilized species) category of useful plant species [7]. NUFCs (neglected and underutilized food crops) exhibit great nutritional value; however, their contribution in ensuring nutrition security is poorly recognized, and they are not included in countries' diet and nutrition programs and policies [8]. Pumpkins (*Cucurbita* spp.) are one of the most overlooked and underutilized food and medicinal plants. Its production is hindered by a scarcity of genetically improved seeds [9]. It is an extraordinary vegetable with the potential to be used as a medicinal as well as a nutritious multifunctional food. Pumpkin peels and flesh contain essential minerals as well as phytochemicals (β-carotene, total flavonoids, and total phenolic) that can help with antiaging and the immune system. Particularly, pumpkin seeds are rich in zinc, and in the midst of the COVID-19 epidemic, scientists are fully aware of the oxidizing as well as mediating effect of zinc with stimulation of enzymes in the body. Since, low-cost powders made from pumpkin components could be employed in the food and pharmaceutical industries as a source of functional ingredients and nutraceuticals [10], pumpkin paste can be manufactured ahead of time and frozen for later use. It has the potential to improve people's nutritional status, particularly among vulnerable groups, in terms of vitamin A requirements. Night-blindness is a serious problem in South Asian countries; promoting the wider peoples to eat more pumpkins can easily remedy the problem [11]. Pumpkin consumption can be utilized to enhance vitamin A intake, fortify diets, and diversify CFs by including animal-derived foods [12]. As indicated by new research interests concentrating on fiber utilization in the modification of the glycemic response and therefore also diabetes, these phytochemicals were found to directly impact the nutritive food quality examined by the usage of pumpkin derivatives as a diabetes treatment in Asia [13]. In Canada and the United States, pumpkin is a popular Halloween staple and thanksgiving food. Pumpkin cultivation is still ignored in most tropical states. It faces a number of challenges, including a shortage of genetically improved seeds, pest infestations, and infections such as the mosaic virus disease and the fungal mild dew [14]. Furthermore, most urban dwellers choose foreign vegetables and regard pumpkins as insignificant crops, reducing the chance of genetic degradation [15]. As a result, there is a growing demand among growers to use hybrids. Hybridization is the greatest beneficial strategy for overcoming the yield obstacle, and it has been employed most efficiently in maize production in the United States. Resistance to diseases, larger adaptability, improved fitness, better yield, and the production of superior plants species are only a few of the main benefits of hybridization [16]. Hybrid ($F_1$) produces 20–30% more yield than inbred or open fertilized variety [17]. As a result, hybrid breeding is preferred over inbred-line mating for a variety of reasons, including enhanced yield consistency and hybrid hetero-zygosity, which allows for the mixing of dominant genes and hybrids, resulting in a developed crop variety conservation strategy [18]. Thus, a hybrid is formed by mating two distinct varieties of the identical plant and may develop cultivars with superior qualities relying on importable attributes to enhance yield, average weight of fruits, pests, disease resistance, environmental adjustment difficulties, enhanced shelf lives, or more mineral concentration. This method also included screening, genotyping based on germplasm assets' genetic diversity, and hybridization through a set of crossings relying on nutritional-related targets. The aim is to produce diversity and a superior performance than its parents for desirable qualities. Before launching variety, appropriate parental strains for the establishment of hybrids with improved qualitative features and attributes must be identified or screened in numerous environmental tests [19,20]. As such, the major

goals of pumpkin (*Cucurbita* spp.) breeding programs are higher yield, pest resistance, and superior quality of young and matured fruit [21] and for increasing nutrient content in crops to generate hybrid varieties and manipulate the heterosis concept [22]. Identifying the diversity of pumpkin genotypes is necessary for conservation, appropriate utilization, and development [23]. Plant genetic diversity allows plant breeders to create new and better varieties with desired characteristics [24], thus determining the extent of variety of qualitative as well as quantitative attributes available in genetic resources is critical for breeding programs [25]. The breeder can choose the correct sort of parents for a hybridized plan by collecting germplasm and assessing the nature and extent of genetic diversity in a disciplined method [26]. The diallel cross mating design provides information on genetic mechanisms established and exemplified [27,28]. It measures parents combining ability and one of the strongest tools in detecting the best combiners that may be used to exploit heterosis or hybrid vigor [29] or to gather fixable genes [30]. Typically, pumpkin has higher content of β-carotin, which is a structural component of vitamin A. However, consumption of this fruit-vegetable provides essential nutrients such as vitamin A, lutein, and zeaxanthin for the people of low-income countries in world. Despite low prices, high nutritional profiles, and diverse potential uses, people as well as researchers have not given proper attention to enhance and popularize this fruit-vegetable. Global improvement of this crop is now crucial for its production, consumption, and multiple uses for general consumers and growers. In view of this point, in this review we add some recent findings of the authors' research) regarding crop improvement. We hope this recent finding will be beneficial for pumpkin breeders and academics working on future breeding programs. Moreover, this study will provide up to date information on how to generate high-yielding hybrids based on knowledge of pumpkins combining ability, gene action, and heterotic potential.

## 2. Botanical Description of Pumpkin

Pumpkins (2n = 2x = 40) belonging to the *Cucurbitaceae* family of flowering plants and genus Cucurbita are among the largest vegetable crops [31]. The *Cucurbitaceae* family comprises eight tribes, 118 genera, and 825 species, including pumpkin [32]. It is one of the morphologically most varied genera in the plant kingdom [33]. *Cucurbita pepo* L., *Cucurbita maxima* Duchesne, and *Cucurbita moschata* Duchesne are three economically important species with a wide range of climate adjustments and broadly circulated to worldwide agricultural regions [34]. Pumpkin is a day-long climbing or creeping plant with a stem that can grow to be more than 10 metres long, an annual, monoecious, and short-lived perennial root. The male flower is distinct from female ones (Figure 1) based on color, structure, and growth. It is seen to be that the female flowers are more elongated (6–12 cm) than the male flowers (3–5 cm) and the color of both flowers varies from yellow to pale orange [35]. Their distribution is influenced by water availability, precipitation, irrigation, and local feeding habits [36]. The Cucurbitaceous family seems to have a wide range of reproductive systems. Monoecious reproduction is the most prevalent reproductive system, where both the male and female sexual components of bloom are generated on the same vine [37]. On the same plant, Cucurbitaceae have both hermaphrodite (complete) as well as staminate (female or male sterile) blooms [37]. Because only the female portion of a flower is developed, gynoecious is beneficial for hybrid seeds [38].

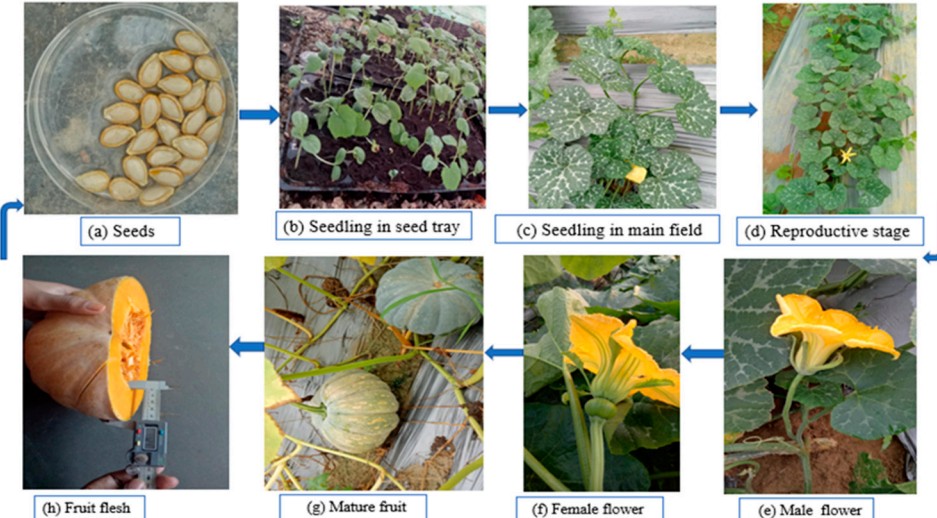

**Figure 1.** Different growth and developmental stage of pumpkin. Note: (**a**) Seed, (**b**) Seedling in tray, (**c**) Seed ling at field condition, (**d**) Reproductive stage, (**e**) Male flower, (**f**) Female flower, (**g**) Mature fruits, and (**h**) fruits flesh.

## 3. Origin and Distribution of Pumpkin

Cucurbita is native to Latin America [36] and has been widely cultivated for over 10,000 years in North America and more than 500 years in Europe [39]. Pumpkin was introduced to Chile and Argentina before finally spreading to Asia and Europe. Irrespective of altitude, they are cultured in almost all parts of the world [35]. Pumpkin is a Western hemisphere fruit-vegetable with several varieties found in North America, Continental Europe, New Zealand, Australia, and India. Pumpkin has also been grown in Malaysia, the Philippines, and Indonesia under tropical Asia [40]. In addition, some countries, including the Russian Federation (accessions-2064), Spain (accessions-925), Germany (accessions-857), Czech Republic (accessions-753), and Hungary (accessions-732), have extensive collections of Cucurbita germplasm [21].

## 4. World Scenario of Pumpkin Production and Area Harvested

Pumpkins (*Cucurbita* spp.) are a major global economic crop in the Cucurbita genus. They are a common food source and an essential source of cultivar for Cucurbitaceae [41]. Pumpkin plants are best cultivated in good, well-drained soil during the warm season. The pumpkin fruits are very susceptible to heat and highly unpreserved in warm conditions [36]. The pumpkin and squash are grown on a global scale of around 3 million hectares, yielding 27.832 million tons. China leads the global pumpkin production with about 58% produced, while India 20%, Ukraine 4%, and Russia 4% are also significant producers. Turkey is in tenth place with roughly 0.6 million tons, accounting for 2 percent of global production (Figure 2). Furthermore, Guyana had the highest yield (kg) per hectare, while the Republic of Korea ranked tenth (Figure 3) and China covered the world's largest pumpkin producing region (Figure 4) [42]. According to FAOSTSAT [42], due to low production of pumpkin, there is an opportunity to spread this crop cultivation and utilization in the tropical regions such as Mali, Thailand, Malawi, Malaysia, Djibouti, Barbados, Antigua and Barbuda, and Dominica.

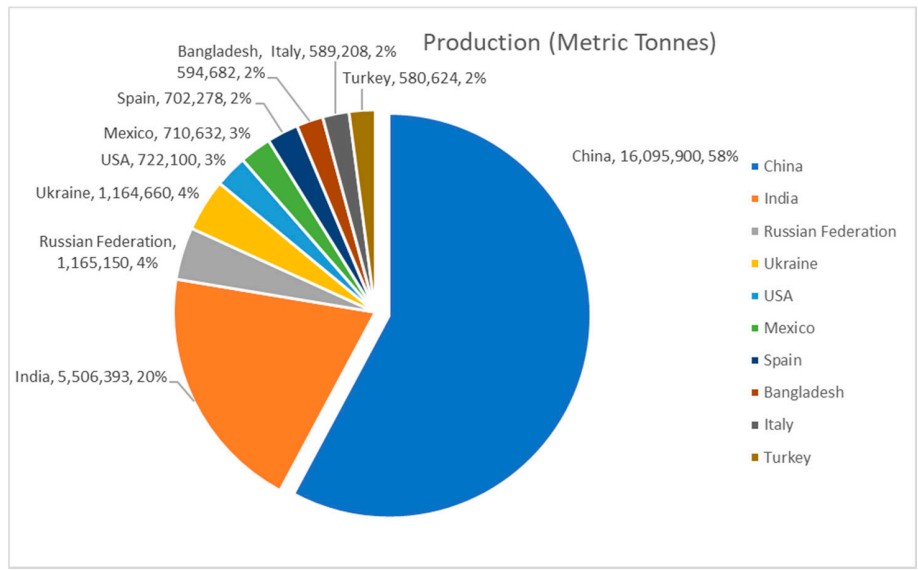

**Figure 2.** Pie chart showing top 10 (ten) pumpkin producing countries and their contributions [42].

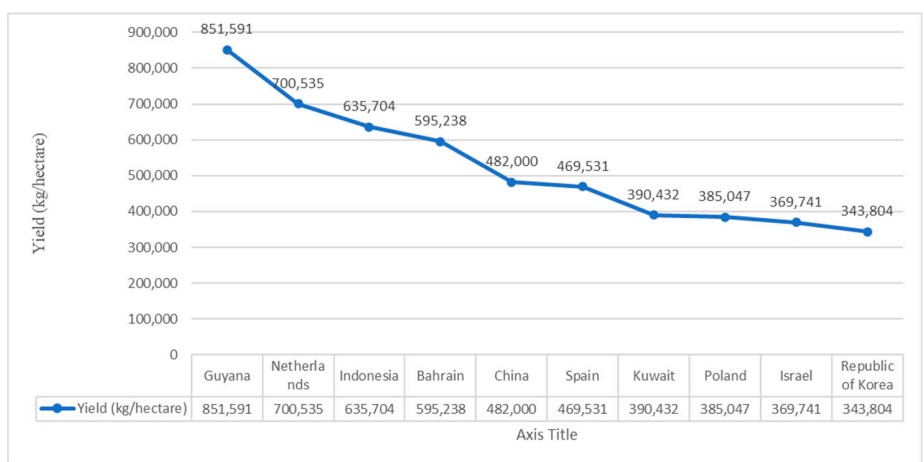

**Figure 3.** Line graph showing top 10 (ten) countries for pumpkin yield (kg/hectare) [42].

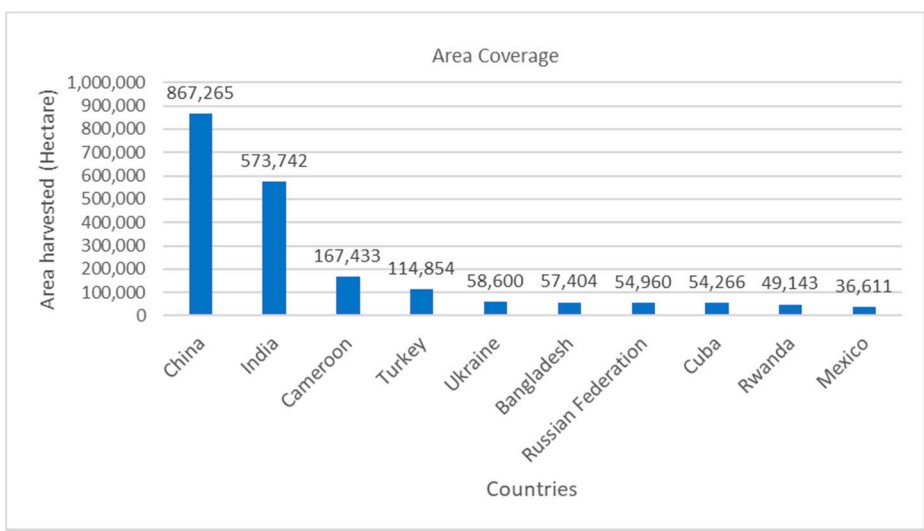

**Figure 4.** Bar graph showing top 10 (ten) countries for pumpkin producing area (hectare) [42].

## 5. Nutritional Aspects of Pumpkin Parts (Leaves, Pulp, Seeds, Seeds Oil, Powder)

Pumpkin leaves (*Cucurbita* spp.) are widely consumed in Sub-Saharan Africa, Asia, Korea, the Pacific Islands, India, and Bangladesh, and are considered nutritious and functional vegetables [43,44]. Pumpkin pulp is high in vitamins, antioxidants such as carotenoids, lutein, zeaxanthins, and minerals (major and minor minerals), making it suitable for human consumption [35,45]. The pumpkin pulp, peel, and seeds are the richest sources of phytochemicals such as polyphenols and flavonoid content, and have antioxidant activity [46]. Pumpkin is considered a valuable vegetable with medicinal and functional culinary properties [10]. Pumpkin seeds are used as a component of bread, salami, sausage, mayonnaise, and many other food products [47], and high-quality oil, as well as being a good source of protein, essential fatty acids (omega-3 (w-3) and omega-6 (w-6) fatty acids), and dietary fiber [22,48]. Pumpkin cultivation primarily depends on its edible oil in advanced countries like Hungary, the Czech Republic, Italy, and Spain, and oil content influences by nature of genotypes and growth circumstances [49]. Their bioactive component composition gives them various advantages that are favorable to human health [50]. Pumpkin pulp and seed powder contain a lot of fiber, minerals (Zn), phytochemicals, and beta-carotene [10,51]. Biscuits made with whole wheat flour were shown to be organoleptically acceptable after 60 days of storage, while pumpkin powder cookies were acceptable after 75 days [52].

## 6. Medication and Pharmacological Values of Pumpkin

Plant products have been used as medicine in human eating habits for millennia. Pumpkin's popularity as a food and treatment in traditional medicine for a variety of ailments (anti-macular dystrophy, antidiabetic, hypertension, anticancer, immunomodular, antibacterial, antihyperon, cholesterolemic, intestinal antiparasitic, digestive antiparasitic, anti-inflammatory, analgesic) has gained the attention of several studies [35,53]. *Cucurbita ficifolia* is an important species of the Cucurbitaceae family and it is the natural source of insulin mediator, namely, D-chiro-inositol (D-CI), which acts as an anti-hyperglycaemic to reduce the blood glucose concentration in Type-2 diabetic patients [54]. As a vegetable, pumpkin is becoming a helpful component of shrinking diets due to their low nutritional value (15–25 kcal per 100 g) and the availability of easy-to-digest elements [55]. Pumpkin fruit has abundant health reimbursements, which are summarized in Figure 5.

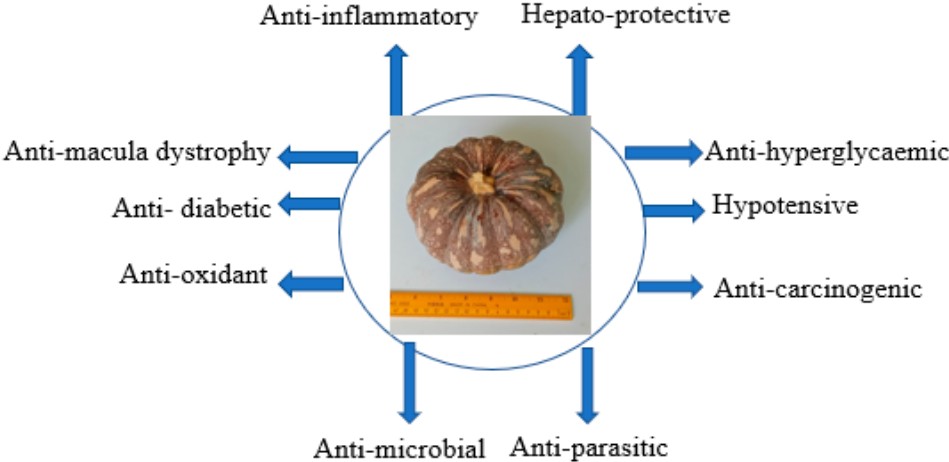

**Figure 5.** Medicinal benefits of pumpkin fruits.

## 7. Potential Uses of Pumpkin (Ready-to-Eat Meals)

Consumers are becoming interested in ready-to-eat (RTE) meals, owing to their ease of storage and preparation, and consumer desirability elements like convenience, cost, attractive look, and texture [56]. According to the Food Standards Agency of the United Kingdom (FSA), ready-to-eat items are defined as "any food for consumption without

any more cooking or preparing." This term refers to both open as well as pre-wrapped ready-to-eat products, and that it applies whether the ready-to-eat meal is hot or cold. Food preparation operations like light cleansing, slicing, cutting, portioning, marinating, even storage carried out by the customer, are not included in the term "additional warming or cooking" [57]. Therefore, processing fruits of pumpkins to foodstuffs that are very easy to carry and stock, such as juices, jelly, purees, jams, pickles, and also dried foods, might overcome these difficulties and increase the marketable value of pumpkin fruits [58]. Pumpkins (*Cucurbita moschata* Duchesne) that are cultivated for their seeds and used as a snack seems to be an emerging trade [13]. Recently, the markets for ready-to-eat (RTE) foods like dried and roasted pumpkin seeds have received a lot of attention [59]. Pumpkin products have a naturally sweet and attractive flavor, and adding pumpkins to dishes enhances beta-carotene quantities. As the snack market expands, manufacturers have been encouraged to develop new products that incorporate a variety of components in order to improve the appearance and nutritional value of their products [60]. The color, flavor, texture, and the nutritional value of fresh-cut vegetable and fruit goods are features that are important to customers [61]. Various ways to use pumpkins as well as potential value-added opportunities are shown in Figure 6.

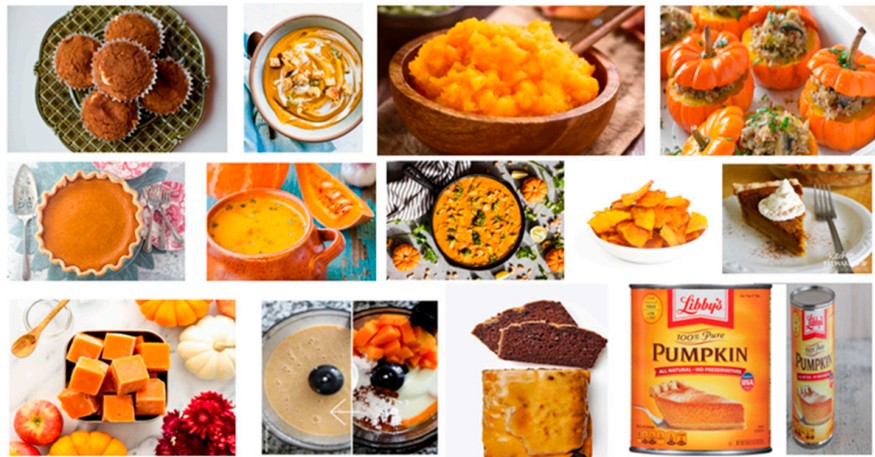

**Figure 6.** Potential uses of pumpkin [62].

## 8. Study of Genetic Diversity in Pumpkin (*Cucurbita* spp.) Genotypes

The effectiveness of a crop breeding program is determined by the genetic diversity availability, genomic data available, and indirect effects on yields and its features [63]. Genetic diversity presence is a precondition for any strategy of crop development. Genetic resources are quite a fundamental keystone of a plan to improve crops [64]. These include the variety of genetic variabilities accessible in landraces, conventional cultivars, assumed ancestral shape, ancient cultivars, types of wild relatives, and associated non-edible wild herbs [65]. This has been realized as a prerequisite for the hybridization breeding of high-yielding cultivars [66]. Many methods have been used to assess crop genetic diversity, including biochemical parameters, morphological attributes, and molecular techniques [67,68].

### 8.1. Qualitative and Quantitative Morphological Diversity of Pumpkin Genotypes

The genetic diversity of germplasms and of *Cucurbita* populations, including shape, size, color variation of fruit, seed number and size, fruit quality, and fruit pulp thickness, are high [69], and therefore can be used in the plant breeding program by developing high yield hybrid varieties. Therefore, a breeder can produce high-yield and high-quality types of varieties by selection [70]. Pumpkins differ in color, size, shape, and weight. It has a moderately hard rind with thick, edible flesh below a central seed cavity [71]. For variety advancement, plant yield and yield contributory traits such as weight of fruit, number of

fruits per plant, length of fruit, diameter of fruit, and weight of 100 seeds, among others, must be considered [72]. Length of fruit (cm), weight of single fruit (kg), TSS (Brix %), and plant yield had the maximum genotypical coefficient of variation [73]. Many researchers agreed that genetic diversity within *Cucurbita* cultivars as well as population groups is large, with variability in fruit shape, color length, as well as pigmentation, relative size of seeds, quality, coloring, width of fruit flesh, pest sensitivity, and fruit precocity, among many other characteristics, [69]. Breeders can construct crop breeding programs using credible information on phenotype diversity within germplasms. The variation in pumpkin fruit colors, shape, and size, leaf architecture, crop stature, lifespan, and the society's constantly changing food patterns have prompted breeders to develop and enhance cultivars in response to demand [74]. To a significant degree, the success of any crop hybrid project depends upon the amount of genetic diversity of the population. Strong attempts are required in the range of higher pumpkin varieties because the current genotypes have broad genetic diversity [75]. It has been suggested that heritability estimates and variants may be used in breeding programs to determine genetic diversity and select better parental types to ensure the cycle's continuation and effectiveness [76]. As a result, predicting these genetic characteristics provides insight into the degree to which a trait can be passed on via subsequent generations of the upgraded species [77]. This projection permits researchers to assess and detect genotype reactions to indices of genetic variance directing a crop development program [78]. The resulting qualitative traits with frequencies (Figures 7 and 8) like leaf color, fruit ribs, fruit shape, fruit skin color, flesh color, and fruit skin texture were observed in the authors' research field and were evaluated as a guideline of cucurbits descriptors [79], whose diverse qualitative characteristics are summarized in Table 1.The high genotypic as well as phenotypic coefficients of variation (Table 2) are important in determining the extent of variation represented in the pumpkin genotypes.

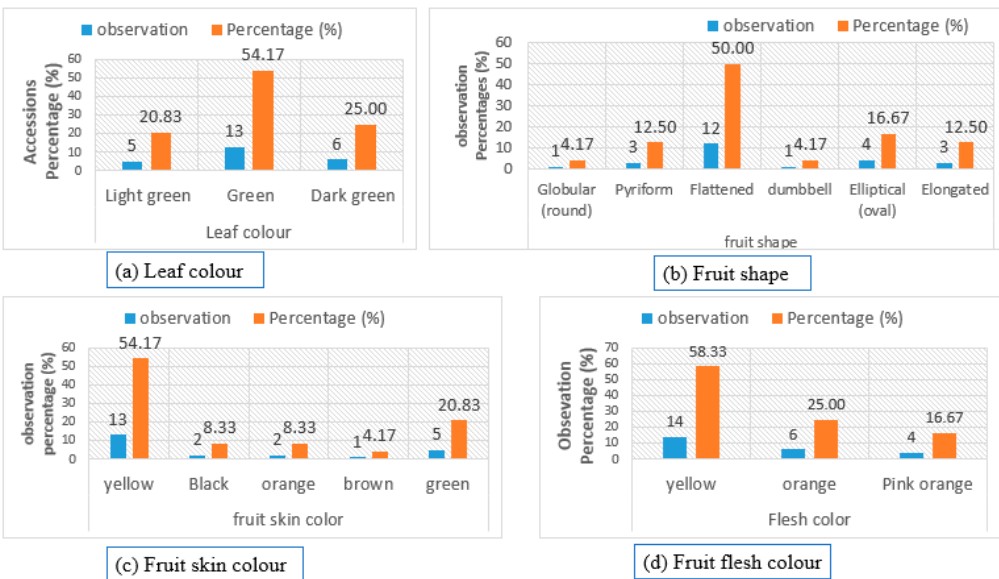

**Figure 7.** Bar graph showing qualitative character variation in (**a**) leaf color, (**b**) fruit shape color, (**c**) fruit skin color, and (**d**) fruit flesh color of different pumpkin genotypes from authors research field.

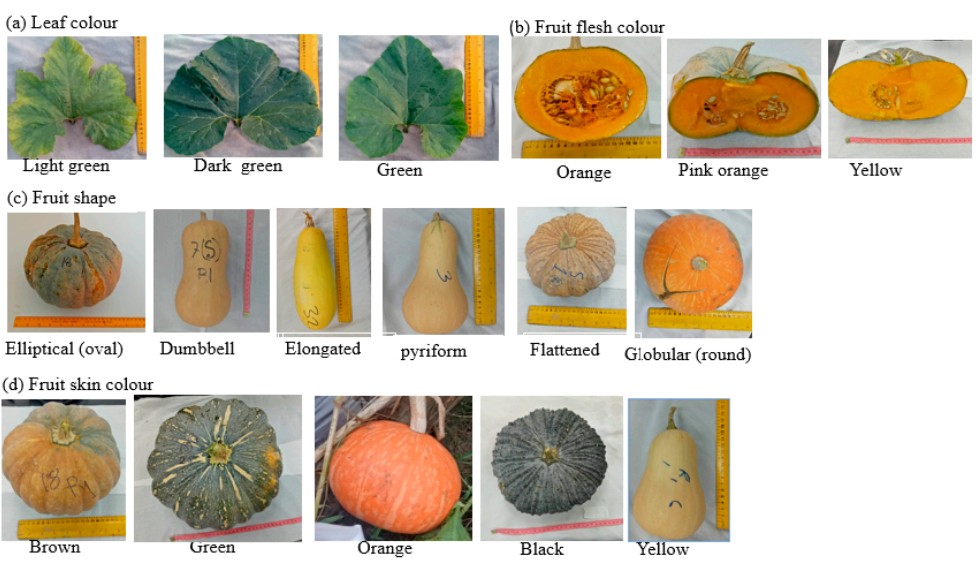

**Figure 8.** Variation in (**a**) leaf color, (**b**) fruit flesh color, (**c**) fruit shape, and (**d**) fruit skin color of different pumpkin genotypes, picture captured from author's research field.

**Table 1.** Reported major qualitative characters variation of pumpkin genotypes.

| Traits | Scale | Reporters |
|---|---|---|
| Leaf colour | Light green | [80] |
| | Green | [80] |
| | Dark green | [80] |
| Fruit shape | Globular (round) | [81–83] |
| | Pyriform | [82,83] |
| | Flattened | [81,83] |
| | Dumbbell | [81,83] |
| | Elliptical (oval) | [82,83] |
| | Elongated | [80] |
| Fruit ribs | Deep | [80,82] |
| | Intermediate | [80,82] |
| | Absent | [80,82] |
| | Superficial | [80,82] |
| Fruit skin colour | Yellow | [80,83] |
| | Black | [80,83] |
| | Orange | [80,82,83] |
| | Brown | [80,82,83] |
| | Green | [80,82,83] |
| Flesh colour | Yellow | [80,83] |
| | Orange | [80,82,83] |
| | Pink orange | [80,82,83] |
| Fruit skin texture | Smooth | [80,83] |
| | Grainy | [80,83] |
| | Shallowly wavy | [80,83] |
| | Finely wrinkled | [80,83] |

**Table 2.** The high genotypic and phenotypic coefficient of variation for diverse characters of pumpkin genotypes.

| Research Materials | High GCV and High PCV Observed Variables | Researchers |
| --- | --- | --- |
| Pumpkin-21 genotypes | Length of vine, fruits number per plant, plant yield, reducing sugars as well as β-carotene | [84] |
| Pumpkins (*Cucurbita moschata* Duch. ex. Poir)-32 genotypes | Plant yield, length of vine, fruits number per plant, weight of fruit, and weight of 100 seeds | [85] |
| *Cucurbita moschata*-91 genotypes | Seed number and seeds mass per fruit, days for blossoming, content of total carotenoid, and production of fruit | [86] |
| Pumpkin genotypes | Total carotenoid's | [87] |
| Pumpkin genotypes-23 | length of vine, cavity of fruit, seed number per fruit, thickness of rind, and fruits number per vine | [88] |
| Pumpkin genotypes-25 | Weight of fruit | [89] |
| Pumpkins (*Cucurbita maxima* Duchesne)-40 genotypes | Fruit yield/plant and content of carotenoids | [90] |
| Pumpkins (*Cucurbita moschata* Duch. Ex Poir.)-19 genotypes | Average weight of fruit, thickness of flesh, fruits number per plant | [91] |
| Pumpkin accessions-7 genotypes | Flesh thickness in the peduncle | [92] |
| Pumpkin-21 genotypes | Length of fruit, weight of single fruit, TSS, and plant yield | [73] |
| Pumpkin-30 genotypes | Days to the first harvesting, ridge number per fruit, diameter of fruit length of fruit average weight of fruit, fruits number per plant, thickness of flesh, diameter of seed cavity, seed number per fruit, and fruit yield per plant | [93] |
| (*Cucurbita moschata* Duch. Ex Poir.)-76 genotypes | Seed yield per plant followed by weight of 100 seeds | [94] |
| Pumpkin-76 genotypes | Weight of plant weight of matured fruit, and yield of fruit per hectare | [95] |
| Pumpkin-40 genotypes | Weight of fruit, plant fruit yield, content of ß-carotene, thickness of flesh, node at the first male flowering, test weight of 100 seeds, fruit equatorial circumferences, and seed number per fruit. | [96] |

*8.2. Molecular Diversity Study in Pumpkin Genotypes*

Expanding the crop's genetic basis, selecting varieties, and choosing parents varieties for breeding programs requires a better understanding of its molecular diversity, pumpkin is poorly characterized in this regard [97]. Because of their greater importance and the fact that they are unaffected by environmental influences such as physical features, molecular markers are selected [98]. In order to evaluate the level of genetic diversity between species of the Cucurbita genus, molecular characterization is needed in addition to morphology characteristics. So many genetic diversities of Cucurbita have been studied in the last couple of decades, along with several types of molecular markers. Many DNA marker systems have produced SSRs composed of repetitive nucleotide motifs of between one and six bases, but which are distributed worldwide and easily available in crop species, are genetically co-dominant, highly reproducible multi-allelic, and ideal for high-throughput genotyping. They are commonly used in plant breeding and plant genetics and produced from random DNA genomic sequence, and expressed sequence tag (EST) markers have been produced and employed in genetic diversity study for various species. Nonetheless, very few investigations have explored genetic diversity with SSR markers in *Cucurbita moschata*, *Cucurbita maxima*, and other related species [99]. Recent advances in molecular genetics have resulted in a tremendous increase in the use of SSRs and SNPs in genetic

studies, making them the strongest options available [100]. With the SSR markers, the researchers attempted to quantify the genetic diversity of *Cucurbita* spp. for appropriate utilization, preservation, and breeding. Table 3 summarizes the various types of molecular markers utilized for genetic diversity studies in *Cucurbita* spp.

**Table 3.** Different types of molecular markers used for genetic diversity study in *Cucurbita* spp.

| Markers Name | Reporters |
| --- | --- |
| RAPD (Random Amplified Polymorphic DNA) | [97,101,102] |
| AFLP (Amplified Fragment length Polymorphism) | [103,104] |
| SRAP (Sequence Related Amplified Polymorphism) | [97,105] |
| SCAR (Sequence Characterized amplified region) | [106] |
| SSRs (Simple Sequence Repeats) | [102,106–108] |
| ISSRs (Inter-simple Sequence Repeats) | [103,109,110] |
| SBAPs (Sequence-Based Amplified Polymorphism) | [97] |
| SNPs (Single Nucleotide Polymorphisms) | [111,112] |

## 9. Hybridization Breeding for Crop Improvement

The two most common sexual forms of hybridization are intergeneric as well as interspecific. Interspecific hybridization occurs when two species cross-fertilize, whereas intergeneric hybridization occurs when two genera cross-fertilize, resulting in an offspring having the phenotypic as well as genotypic characteristics from both parents, encouraging genetic diversity as well as evolvement [113]. Hybrids are the first filial generation (F1) offspring of two different pure lines crossed [114]. Hybridization (Figure 9) is a method of exploiting heterozygosity to improve agricultural performance, attributes, and productivity [115]. The mid-parent heterosis is defined as the difference in productivity between the hybrids and the mean of its parental genotypes. The better parent heterosis [116] is defined as hybrid productivity that is superior to the greater or higher parental line. It is utilized in cultivar development to increase vigor and create more diversity. Hybrids have a 15–25 percent higher yielding potential over natural pollination [18]. The capability to detect or identify the best possible pairing of two or more ancestral genotypes to maximize variation within such a population is required in hybridization to develop improved transgression segregants in a segregated community of hybrids [117]. In the Cucurbitaceae, numerous breeding designs have been employed to assess general combining ability (GCA) as well as specific combining ability (SCA) variation [118]. GCA and SCA are numerical genetics methods for determining the properties of gene action associated in the development of economically appropriate plant attributes [118] and the capability of parents to combine upon hybridization. The over-dominant concept of plant genetic might be used to describe hybrid development [119]. Over-dominant refers to the usage of an F1 hybrid's superior characteristics in comparison to its inbred ancestors as a result of superior allelic relationships at heterogeneous loci that surpass either homogeneous parent [120]. A mixture of genotypic, epigenetic, morphological, and ecological aspects affects heterosis or hybrid vigor [121].

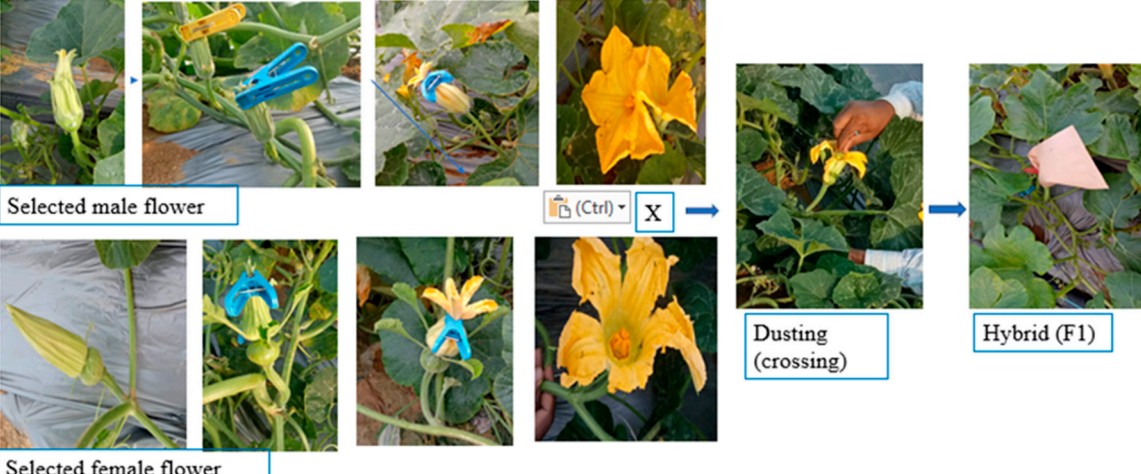

**Figure 9.** Hybridization technique. The desired female and male flowers, which were expected to anthesis at the next morning, were covered with butter paper/cloth clip at the previous evening before the flowers open, and at the following morning the pollens from the selected male flower were dusted on the female flower of the desired plant and covered again till the fruit set to avoid unwanted cross pollination. A label was tagged denoting details of parents and date of pollination.

### 9.1. Diverse Mating Design Used in Hybridization

The prerequisites to the fruitful plant breeding program are the choice of paternal resources and righteous breeding designs in traditional crop breeding. The mating scheme denotes creating the progenies in the crop breeding program [122]. Plant breeders and geneticists use a variety of mating methods and combinations to create superior progenies. Plant breeding solutions rely on determining the appropriate parents and using the right mating patterns [123]. Furthermore, several elements influence mating design choices, which include:

(i)　the mode of fertilization (self- or out-pollinated);
(ii)　the mode of crossing that can be used (non-natural or natural),
(iii)　the mode of pollen transmission (air or insect);
(iv)　the existence of the male sterility structure;
(v)　the objective of the plan (breeding or hereditary analysis),
(vi)　the magnitude of the required population [124].

The mating schemes are important for four reasons:

(1)　they deliver info on the inherited control of the traits under examination;
(2)　they create a breeding progeny that can be used as the foundation for choice and develop probable hybrid varieties;
(3)　they offer estimations of heritable gain; and
(4)　they offer info for assessing the parental materials used in a breeding program [124].

Entities are picked at random and mated through all mating schemes to create progenies connected to one another as half-sibs or full-sibs. A type of multivariate model or ANOVA can be used [122]. Table 4 summarizes the major crossing designs employed in crop breeding programs based on hybridization.

**Table 4.** List of major mating approaches used in hybridization for crop improvement.

| Name of Mating Design | Proposed by | Source of Variance |
|---|---|---|
| Biparental mating design | [125] | Between families, Within Families, [124] |
| Poly cross | [126,127] | Progenies, Blocks [128] |
| Top cross | [129] | Progenies, Blocks [130] |
| Triple cross | [131] | Replication, Genotype [132] |
| North Carolina Mating design-I, (Hierarchical or Nested mating Design) | [133] | Males, Females, Within plots [124] |
| North Carolina Mating design-II, (Factorial mating Design) | [133] | Replications, Males, Females Males × Females. Within progenies [134] |
| North Carolina Mating design- III | [133] | Tester P, Males (F2), m, Tester × Parents Within FS Families [130] |
| Line × Tester | [135] | Replications, Lines, Testers Lines × Testers, Error [126] |
| Diallel mating design. Method-I | [136] | GCA, SCA, Reciprocal effects [136] |
| Diallel mating design, Method-II | [136] | GCA, SCA [136] |
| Diallel mating design, Method-III | [136] | GCA, SCA, Reciprocal effects [136] |
| Diallel mating design, Method-IV | [136] | GCA, SCA [136] |

*9.2. Why Choose Diallel Mating Approach in Hybridization Breeding?*

In the genetic study, diallel crosses have been used to determine the inheritance of important characters among a group of genotypes to identify superior parents for hybrid (F1) or cultivar development. A genetic study of characters needs to be determined to apply a perfect breeding approach for targeted quantitative characters [137,138]. Diallel study of cross and self -pollinated populations is used to measure the genetic control of measurable characters [27,139,140] and to measure specific and general combining abilities [136]. A relative assessment of mating models is summed up in two distinct ways:

(1)  in terms of population coverage, and it stated that Biparental mating > North Carolina I > poly cross > North Carolina III > North Carolina II > diallel mating, in that directive of declining effectiveness;

(2)  in terms of volume of info, diallel mating > North Carolina II > North Carolina III > North Carolina I > Biparental mating, [124].

Although, for GCA as well as SCA, diallel mating approach is the utmost crucial strategy, but as a substitute of trusting on the mating strategy per se, the breeder plays a significant role in mating plan choice. The right selection and use of the breeding design will offer very convenient information for plant breeding [122]. For obtaining crucial genetic information, the diallel has become the most often used and battered mating procedure [130].

The diallel mating design covered the approximation of

-    general combining ability (GCA),
-    specific combining ability (SCA),
-    genetic parameters,
-    heritability in broad and narrow senses,
-    and gene action (additive and non-additive) [141]

*9.3. Research on Development of High Yield and Quality Pumpkin Hybrid Using Diallel Mating Design*

9.3.1. Evaluation of Combining Ability

With the general method of selecting parents due to their individual *per se,* performance may not always get positive results [142]. The study of combining ability analysis preliminarily can be used to detect the merits and demerits of parents and combinations in past generations, permitting breeders to decrease the quantifiable range, save time for breeding, and improve breeding competence [143]. In ecology, all of the ideal plant economic characteristics are hardly seen in a lone entity. As a result, breeders frequently have difficulties choosing parents and their crosses when producing high-yielding cultivars. General combining ability (GCA) analysis enables breeders to pick suitable parents for hybridization and determine the source and degree of gene action in presenting a specific characteristic [144]. However, the specific combining capability (SCA) is described as the deviations from the population average value of the mean results of a certain cross combination. The SCA was controlled by gene action of non-additive, according to [145]. The SCA effect of hybrids has also been related to the mixing of beneficial genes from many sources. An experiment was done in 2016 and 2017 by 6 × 6 half diallel mating design and evaluated results showed GCA calculations were greater than SCA calculations for the most of the plant features considered. All of the traits studied profited by additive and non-additive gene action, according to the research. For height of plant, area of leaf, leaves number per plant, nodes number to first female flowering, and days to female flower anthesis, P1, P2, and P3, are good parents while P5 and P6 are good parents for height of plant, area of leaf, leaves number per plant, nodes number of female flowers, entire plant yield, and TSS [146]. Combing ability were explored in half diallel fashion, including bushy plants and butternut fruits, in order to improve the pumpkin (*Cucurbita moschata* Duch ex Poir.) for intensive farming. P-3621 was the highest overall combiner for equatorial diameter, short vine, internode, and high fruit shape index; P-6242 was best for pulp thickness; and P-2211 represents the number of primary branches, P-10224 represents polar diameter as well as vegetative growth, and P-1343 represents for petiole length [147]. An experiment of 7 × 7 full diallel mating design was done and resulted in the parents P1, P3, P4, P5, and P7 appearing to be the best combiners for average fruit length, shape index of fruit, and fruits number per plant according to the findings. P2 was also the top combiner in terms of total soluble solid percentage (TSS percent). The analysis of variance exhibited highly significant estimates for all components for both GCA and SCA. The GCA was higher than the SCA for all factors. Inbred lines P1, P2, P3, P4, P5, and P6 were the strongest combiners for branch number per plant, length of fruit, days to the anthesis of first female blossom, and fruits per plant, respectively. The inbred strains P1, P2, P3, P4, P5, and P6 were the finest combiners in terms of plant yield, average weight of fruit, leaves number per plant, and stem length [148]. Four parental lines, as well as their six F1 crosses established from a half diallel mating model, as well as three commercial varieties, were studied, and the results revealed that no parent was superior for all of the verified attributes. The father (P3), on the other hand, had the best values for the bulk of the traits assessed, especially earliness and yield [149]. A study revealed that the P3 × P4 F1 crosses had the greatest mean values for length of stem, number of male flowerings per plant, number of female flowerings per plant, average individual fruit weight, and plant yield. The cross P1 × P3 for sex ratio and diameter of fruit, P1 × P2 for length of fruit and shape index of fruit P1 × P4 for fruits number per plant. The three F1 hybrids, P1 × P2, P1 × P4, as well as P3 × P4, showed better mean values for the ratio of sex, length of fruit, shape index of fruit, number of fruits per plant, and yield per plant than the commercial F1 hybrids [149]. ANOVA of the diallel data set also exposed significant and high effects of SCA (specific combining ability) and GCA (general combining ability) between genotypes regarding weight of seed, yield of fruit, the ratio of fruit and weight of seed, seed number of fruits, and seed yield. For fruits number per plant and plant fruit yield, the three genotypes P3, P6, and P7, were shown to be virtuous general combiners [150]. The parents Narendra

Agrim, Kashi Harit, Punjab Samrat, and Pusa Vishwas were recognized as potential donors for earliness and yield characters based on performance and GCA effects [151]. In the parent CM-1, there was evidence of good general combining ability for male and female flowers number per plant, fruit earliness, average fruit diameter, flesh thickness, average fruit weight, number of fruits per plant, plant yield, seed number per fruit, and brix [152]. SCA for most earliness traits was lower than their corresponding estimates of GCA [153]. High specific combining ability (SCA) in P-10224 × P-6242, PS × P-6242, P-41212 × P-2211, PS × P-364, P-41212 × P-6242, P-10224 × P-2211 exposed the existence of both gene effects of additive and non-additive with the majority of non-additive properties ($\sigma2SCA/\sigma2\,GCA > 1$) in the inheritance of butternut and bushy traits, which can be exploited through heterosis breeding and recurrent selection in pumpkin. The hybrids P-3621 P-2211, P-6711 P-2211, and P-6711 P-6242 showed high flesh thickness and yield potential that was superior to the check hybrids, and could be used for profitable cultivation lengths [147]. The hybrid (P1 × P6) provided the highest early production per plant (78.95%) during heterosis over the good parent, followed by a hybrid (P5 × P6) (73.91%) for the same attribute. In terms of overall yield per plant, the hybrid P4 × P6 (75.0 percent) showed the maximum mid-parent heterozygosity, followed by the cross P5 × P6 (60.38 percent) as well as P4 × P5 (59.65 percent). In terms of the overall yield per plant, the cross P4 × P6 (61.77 percent) had the largest heterosis over the better parent, followed by the cross P4 × P5 (46.77 percent) as well as P5 × P6 (37.10 percent), respectively [146]. The dominance of specific and general combining ability for yield and yield attributed diverse characters in inbred and their F1 of different diallel mating approach are summarized in Table 5.

**Table 5.** The dominance of specific and general combining ability resulting of diallel mating design for diverse characters of pumpkin genotypes.

| Sl.no | Research Materials | Traits for Dominance of GCA | Traits for Dominance of SCA | Researchers |
|---|---|---|---|---|
| 1 | 4 inbred lines | Diameter of fruit, flesh thickness, weigh of fruit, weight of seed, and TSS | Fruit yield per plant | [154] |
| 2 | 6 inbred lines | Fruit length, average fruit diameter, thickness of fruit flesh, Fruit weight, quantity of fruits per plant, and total soluble solids in fruits (TSS) | Days to the first female flowering | [155] |
| 3 | 6 inbred lines | Early maturities of fruit, average fruit diameter, individual fruit weight, fruit pulp thickness, number of fruits per plant, total fruit yield, number of seeds per fruit, and characters of seed | Early fruit maturities, average weight of fruit, total fruits number, total fruit yield per plant, characters of seed, and TSS (brix percentages) | [152] |
| 4 | 7 inbred lines and their F1 hybrids | Plant growth speed on the 50th day of transplanting, main plants stem lengths on the 50th day of transplanting, plants productivity, and shape of fruit | Yield per plant | [118] |
| 5 | 5 open pollinated lines | Total fruit output per plant, average fruit weight, and seed weight | Total fruit yield/plant, average fruit weight, 100 seed weight, and number of seeds per fruit are considered. | [156] |

9.3.2. Evaluation of Heterosis or Hybrid Vigor

Heterosis is the superior performance of F1 hybrids (Figure 10) over the superior parent or the parents' mean or the typical check [157,158] depending on the gathering of favorable dominant gene to the F1 hybrid population. Both the female and male parents donated to the gathering of the favorable dominant gene in the F1. In cross-pollinated species, heterosis is more distinct. The studies [159,160] mentioned the dominance of incomplete to complete, epistasis, and overdominance are three probable genetic reasons of heterosis. Heterosis breeding is an essential strategy for agricultural improvement because pumpkin is a cross-pollinated crop. The study [159] discovered heterosis in cucumber for the first time. The genetic study delivers a guideline for calculating the comparative breeding probable of the parents, or otherwise recognized top combiners in the plants that might be applied either to take advantage of hybrid vigor in F1 (hybrid) or to amass fixable genes to develop a variety [161]. All of the characters in all three seasons (E1, E2, and E3) experienced considerable heterobeltiosis and economic heterosis as a result of the hybrids' development. Crosses P1 × P5, P4 × P6, and P1 × P2 may be exploited as commercial hybrids for gainful yield in pumpkin. Significant heterobeltiosis and economic heterosis showed the importance of heterosis breeding for the evolving high yield hybrids (F1) [11]. In a study of 7 × 7 full diallel squash mating with parental genotypes for heterotic exhibition, the genetic behavior of yield, yield attributed, and quality component was explored. A full diallel crosses mating design technique was used to create the 42 F1, 1r hybrids of these parental kinds. The results also showed that the degrees of heterosis vs. mid parents had really significant values for the majority of the variables studied, and the projections of heterosis vs. the better parent were highly significant for the majority of the variables studied. Although some hybrids showed significant heterosis for some variables, none of the hybrids demonstrated increased heterosis for all variables and preferable heterosis over the mid and better parents [162]. Heterosis breeding allows for a one-generation enhancement in yield and other desirable traits that would be more time consuming and difficult to achieve with other traditional breeding procedures [163]. For length of stem, branch number per plant, leave number per plant, days to the first female blossom opening, weight of average fruit, fruits number per plants, plant yield, average fruit length, and average fruit diameter, the maximum significant desirable heterosis over mid-parent values were −17.19, 61, 44.14, −5.46, 17.26, 308.83, 296.41, −23.58, and −12.79%, respectively. Also, for the aforementioned attributes described earlier, the heterosis over better parent was reported at −18.18, 32.47, 37.66, −6.97, 14.77, 205.54, 204.98, −32.63, and −14.48 % [148]. A result exhibited that the uppermost noteworthy accurate heterosis in required volume (179.9%) was noted for early plant yield followed by entire yield (106.9%), number of fruits per plant (57.0%), height of plant (40.9%), weight of average fruit (32.5%), and female flowering days (−17.2%) [164]. Abd El-Hadi et al. 2014 [162] showed that mid-parents heterosis was notable for all studied considered characters, while extents of the better parent heterosis exhibited noteworthy high volumes for most considered characters. Hussien and Hamed, 2015 [164] directed that the uppermost heterosis over superior parents in the required way was noted for early plant yield (179.9%), overall yield plant (106.9%), fruits number per plant (57.0%), as well as height of plant (40.9%). Promising heterosis over better parent observed for weight of fruit, 100 seeds weight, and fruit yield noticed by [165]. The heterotic effects were impressively considerable, and promising hybrids in terms of measurable traits might be obtained. For yield characters, the summer squash showed the greatest heterotic effect. The yield showed high heterosis (32.67%) and heterobeltiosis (24.36%) [166].

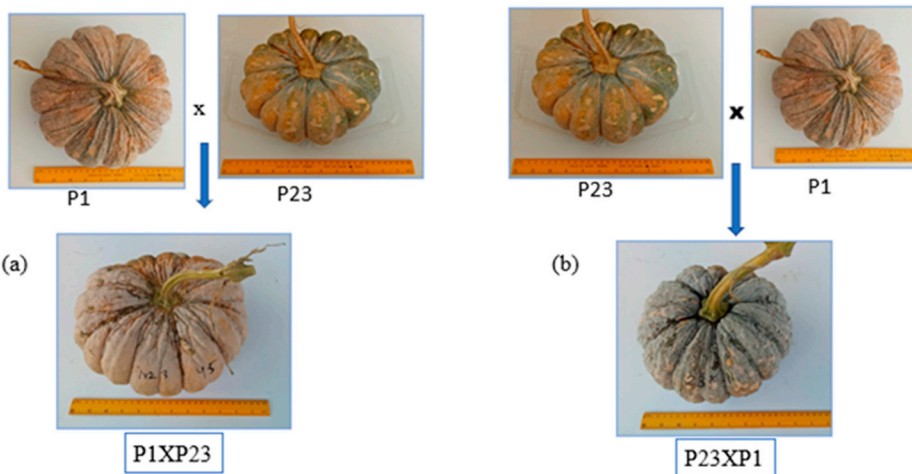

**Figure 10.** Some elite crosses showed more superior performance (hybrid vigor) than parental line. (**a**) Cross: P1 × P23 = hybrid: P1P23), (**b**) Cross: P23 × P1 = hybrid: P23P1.

### 9.3.3. Evaluation of Heritability of the Desired Traits for Pumpkin Breeding

Heritability is frequently assessed in plant breeding (i) as a measure of trial precision and (ii) to quantify the response to selection [167]. The genetic breeding that would result from selecting the best plants is not indicated by heritability estimates alone. Combining heritability with phenotypic variance and selection intensity, on the other hand, allows for the determination of genetic progress or selection responsiveness, which is more beneficial in selecting promising lines [168]. Heritability ($h^2$) is an approximate measure of the expression of a character [169]. For all attributes, heritability estimates in the broad sense ($h^2b.\%$) were greater than those in the narrow sense ($h^2n.\%$) [148]. The node number of females blossoming and TSS ranged from 72.95 to 99.46 percent, respectively. In addition, for the leaves number of plant and the index of fruit form, heritability in the partial sense fluctuated from 28.49 to 81.94 percent [146]. The broad-sense heritability estimates ($h^2b\%$) were higher and more significant than the narrow-sense heritability estimates ($h^2npercent$). However, narrow-sense heredity estimations for earliness ranged from 4.02 percent to 13.46 percent, whereas sex ratio (SR) and branch number per plant inheritance measurements were also between 4.02 percent and 13.46 percent. For total yield and average weight of fruit per plant (g), narrow sense heredity estimates varied from 8.92 to 18.39 percent, respectively, for yield components. Eskandarany (P1) was discovered to be exceptional overall combiners for fruit earliness and yield associated characteristics [170]. The heritability of few traits associated to fruit quality and yield in *Cucurbita moschata* with 7 × 7 half-diallel crossing design. Highest heritability value was observed for the beta-carotene (98.6%), for fruit weight (82.9%), and second highest value for seed size index (89.5%). For fruit number of plants, fruit yield, and individual fruit weight, researchers discovered a high genetic advance fixed with high heritability in parents and crossings, suggesting that development would be actualized via phenotypic choice [171]. The highest value was observed in case of broad ($h^2 b$) as well as narrow ($h^2 n$) sense heritability for seed yield, total fruit yield, weight of seed, ratio of fruit weight to seed weight and seed number per fruit, while average fruit weight and 100 seeds weight showed low value for both of heritability [165]. Furthermore, the assessments of narrow-sense heritability were lower in amounts than their corresponding broad-sense heritability values. The highest value (57.196%) of the pooled data was observed for the shape index of fruit attribute. At a similar time, the heritability value in a broad sense varied from 99.224 to 99.762% for average fruit length and average fruit diameter for the pooled data [154]. For all calculated characters, heritability in narrow-sense ($h^2n$) assessments was lower in value than in broad sense ($h^2b$). Broad-sense heritability for yield and yield component traits ranged from 0.513 percent to 0.894 percent [172].

### 9.3.4. Evaluation of Gene Action of the Desired Traits for Pumpkin Breeding

Studies on gene action reveal the genetic mechanism and the inheritance of quantitative traits, specifically with gene actions with the effect of epistasis as well as additive gene, the direction of dominance in the parents, dominance degree, allocation of negative and positive alleles as well as their linkage, and estimation of specific and general combining abilities of the parental lines involved. Among the many methodologies, genetic analysis provides a viable approach for evaluating newly established varieties for their parental efficiency and measuring the gene action involved in numerous qualities to develop an appropriate breeding plan for extra genetic advancement of the existing material [136]. The consequences of the diallel mating design study exposed that (H/D)1/2 value was more than unity for all calculated characters representing over-dominance. Though, the preponderance of nonadditive gene effects and values of heritability in narrow sense for most important characters express the opportunity to exploit dominance gene effects for improving such characters over heterosis breeding [164]. In an effective crop breeding program, the knowledge about the relative role of variance components, viz., nonadditive and additive, is important [157]. Research conducted by $6 \times 6$ diallel crossing avoiding reciprocals with four foreign cultivars and two local cultivars to identify gene action for some economic attributes in squash. The genotypes mean squares and their components, general combining ability (GCA), and specific combining ability (SCA) were highly significant for all examined features, showing both additive and non-additive genetic variance influenced to the inheritance of the analyzed attributes. For all of the parameters tested, both GCA and SCA genetic variations were shown to be extremely significant, viewing the role of additive and non-additive genetic activity [172]. The additive genetic variance ($\delta^2$A) was lower than their corresponding of dominance ($\delta^2$D) heritable variance values of for the trait of the node number at first female anthesis [153]. The predominance of additive gene action was observed for entire characters excepting weight of 100 seeds and first female flowering node, while nonadditive gene action was predominant [173]. The days for first female flowering had higher SCA value than GCA variance representing the predominance of nonadditive gene effects [155]. For the traits of nodes at first female anthesis, the ratio of GCA and SCA was less than one that showed the multitude of dominance expression [174]. Both dominance and additive genetic variance were noticed significant in most of the traits' inheritance, although dominance variance was more protuberant than the variance of additive in all the seasons [175]. All yield and yield representative features were positive for both additive and nonadditive genes, with the exception of additive heritable variation in flesh thickness, fruit diameter, fruit weight, and seed weight [154]. In genetic parameters estimation, the additive genetic variance was very important for most considered characters. Additive heritable variance instead of nonadditive as well as cytoplasmic inherent factors governed the inheritance of measured characters [153].

### 10. Future Prospects

Hybridization breeding is oriented on the morphological expression of the plant rather than genotypic characteristics. The procedure is time-consuming, labor-intensive, restricted to closely connected species, and may transmit undesirable genes with desirable genes. Furthermore, traditional breeding may be hampered by genetic erosion, genetic drag, and reproductive difficulties. In addition, due to the introduction of novel biotypes, and strains of insect-pests and abiotic challenges caused by climate change as well as the worldwide demand for nutritious food, breeding aims and methods have reoriented to meet the world's future demands. With the expanding population, dwindling inputs (land, water), as well as the visible going lower of yield curves, traditional breeding alone will no longer be able to meet world food demand. As a result, combining traditional and novel plant breeding techniques such as bioinformatics, molecular biology and genetics, genetic engineering, bioengineering, mutation breeding, MAS (Marker Assisted Breeding), genetic transformation, and tissue culture is required for crop development in the future. Consid-

ering the overall incidence, recommendations for pumpkin improvement, popularization, its spread and cultivation in tropical area, as well as global aspects are outlined here:

1.  More awareness needs to be spread to tropical populations about the need of eating fruits and vegetables, with pumpkin being one of them, as one of the measures to promote human health and combat hunger.
2.  Youths as well as women must be encouraged and informed about pumpkin production as well as the various value-added options for pumpkin that can provide potential sources of revenue and contribute to overall socioeconomic prosperity. Farmers should be motivated to keep track of their production in order to monitor their agribusiness's success.
3.  In case of surplus pumpkin production, this expertise of pumpkin value addition should be disseminated with local people to reduce postharvest wastage, which would benefit the overall socio-economic well-being of the growers concerned.
4.  The tropics' pumpkin landraces require the establishment of a GenBank for maintenance and future crop enhancement to address yield as well as pests and illnesses. Landraces collecting and in situ preservation aid to combat the threat of endangerment and loss of crop genetic variation.
5.  Local species' genetic variability can be exploited to generate superior pumpkin cultivars that can withstand changing and limited environmental conditions. The introduction of foreign cultivars with a confined genetic background has put the genetic variability and preservation of landrace pumpkins at risk.
6.  A cost-benefit economic analysis of value-added options should be conducted to identify which option will provide farmers with the highest profit margin for their pumpkin harvest.
7.  Governments, community, entrepreneurs, and other stakeholders must work together to encourage the popularization of native landraces by generating high-quality seeds, creating new disease-resistant, high-yielding varieties, and supporting traditional food festivals to encourage indigenous cuisines.
8.  Nutrient analysis and therapeutic applications of naturalized pumpkins are needed. Traditional food manufacturing techniques must be investigated in order to help local people in terms of medicine, nutrition, and economics.

## 11. Conclusions

The unrecognized crop pumpkin has the potential to be effective in achieving national and global food and nutritional security, while current main crops are badly fitted to the environment. Its genetic breeding in tropical and subtropical climates is also delimited. Despite the fact that pumpkin plants are often ignored, the implementation of contemporary breeding technologies for this crop provides a lot of potential benefits. For the breeding of pumpkin genotypes regarding the yield and quality traits, the hybridization breeding through the exploitation of combining ability, gene action, and heterosis vigor using diallel mating design might be a solution to mitigate the food and nutritional challenges.

**Author Contributions:** Formal analysis, M.H., M.M.H.K.; Funding acquisition, M.Y.R.; Investigation, M.Y.R., N.M. and M.J.; Supervision, M.Y.R.; Validation, M.Y.R.; Writing—original draft, M.H.; Writing—review & editing, M.H., M.F.N.C., M.M.H.K., I.M. and Y.O. All authors have read and agreed to the published version of the manuscript. All authors offered suggestions on various drafts of the manuscript.

**Funding:** This research was funded by Universiti Putra Malaysia (UPM), Malaysia.

**Institutional Review Board Statement:** Not applicable.

**Informed Consent Statement:** Not applicable.

**Data Availability Statement:** All data is available in the body of the manuscript.

**Acknowledgments:** The authors are thankful to the Ministry of Agriculture (MoA), Bangabandhu science and technology fellowship trust of the People's Republic of Bangladesh and Universiti Putra Malaysia (UPM).

**Conflicts of Interest:** The authors declare no conflict of interest.

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
