# Peer review of "Pumpkin (Cucurbita spp.): A Crop to Mitigate Food and Nutritional Challenges"

_horticulturae, doi:10.3390/horticulturae7100352_

Round 1

Reviewer 1 Report

  1. I suggest the authors change the title of the paper which is too long and too complicated. The title of a paper cannot capture all the aspects contained in that paper.
  2. In Key words, I consider the word "improvement" unsuggestive for this paper.
  3. There is a slight confusion about the authors' own results and those in the literature. Example- figure 1, 9, 12 and related comments. Therefore, it is necessary to clearly mention at the end of the INTRODUCTION chapter the weight of the two components and the way in which this paper is composed from this point of view.
  4. The INTRODUCTION mentions “So the major goals of pumpkin (Cucurbita spp.) Improvement programs are higher yield, pest resistance, and superior quality of young and matured fruit [21]”. I ask the authors to mention which pests it refers to, as it is known that the pumpkin does not have pests of economic importance like other vegetable species.
  5. In the INTRODUCTION the authors write: "Identifying the diversity of pumpkin genotypes is necessary for conservation, appropriate utilization, and development [23]." This idea is found throughout the work. My question is whether genetic diversity in pumpkin has been sufficiently studied and it has been concluded that existing varieties cannot cope with the goal of extending this species to the tropics? It would be interesting to specify the deficiencies of the existing varieties and landraces. Variety breeding is very expensive and long lasting.It would also be interesting for the authors to mention the reasons they identified for not cultivating pumpkins in the tropics.       
  6. Chapter 3. "Origin and distribution of  pumpkin" mentions "A couple of countries have wideranging collections of Cucurbita germplasm [21]." Even if the figure is very explicit, I suggest that at least the first 3-4 countries be listed in the text.
  7. The "Introduction" states "Pumpkin cultivation is still ignored in most tropical states." Chapter 4 “World scenario of pumpkin production and harvested area” states: “Pumpkin plants are best cultivated in good, well-drained soil during the warm season. The pumpkin fruits are very susceptible to heat and highly unpreserved in warm conditions [36]. ” The paper would need to outline the area where the pumpkin could spread to tropical states. The idea must also continue with recommendations on how to spread this crop. To whose detriment do the authors recommend that pumpkins be grown in the tropics? Such recommendations, scientifically documented from all points of view (economic, ecological, nutritional, etc.) could convince the leaders of tropical states to introduce pumpkin in their future agricultural programs.

Author Response

Ref: horticulturae-1353785

MS TITLE: “PUMPKIN (Cucurbita spp.): A CROP TO MITIGATE FOOD AND NUTRITIONAL CHALLENGES

Thank you very much for your valuable criticism and suggestion regarding this manuscript. As per reviewer comments and suggestion the manuscript has been revised properly through track change in the main text of the manuscript as well as the point-to-point response of reviewer 2 is given bellow.

Point 1: The effort made by the authors is very valuable, as nowadays it is impossible to follow most of the current literature on a topic of interest. This paper is a comprehensive review focused on the potential uses of pumpkins. The manuscript fits within the scope of the journal. The manuscript is interesting and the idea is nice. The title is clear and it is adequate to the content of the article. The author’s work on discussing achieved results is appreciated. Minor revisions are necessary to improve the clarity of the presentation.

Response; Thanks for your critical assessment.

  1. I suggest the authors change the title of the paper which is too long and too complicated. The title of a paper cannot capture all the aspects contained in that paper.

Response: The title of the paper has been changed “Pumpkin (Cucurbita spp.): A crop to mitigate food and nutritional challenges”

  1. In Key words, I consider the word "improvement" unsuggestive for this paper.

Response: The unsuggestive word "improvement" in the Key words is deleted.

  1. There is a slight confusion about the authors' own results and those in the literature. Example figure 1, 9, 12 and related comments. Therefore, it is necessary to clearly mention at the end of the INTRODUCTION chapter the weight of the two components and the way in which this paper is composed from this point of view.

Response:  The justification of the addition of author’s research findings in manuscript literature is stated at the end of the INTRODUCTION section as per your comments.

  1. The INTRODUCTION mentions “So the major goals of pumpkin (Cucurbita spp.) Improvement programs are higher yield, pest resistance, and superior quality of young and matured fruit [21]”. I ask the authors to mention which pests it refers to, as it is known that the pumpkin does not have pests of economic importance like other vegetable species.

Response: The pests we are referring to here are linked to red pumpkin beetle (Aulacophora foveicollis), fruit fly (Bactocera cucurbitae, Bactocera dorsalis) and white flies (Bemisia tabaci) which is the agents of viral infections transmits particularly as observed in the authors research field.

  1. In the INTRODUCTION the authors write: "Identifying the diversity of pumpkin genotypes is necessary for conservation, appropriate utilization, and development [23]." This idea is found throughout the work. My question is whether genetic diversity in pumpkin has been sufficiently studied and it has been concluded that existing varieties cannot cope with the goal of extending this species to the tropics? It would be interesting to specify the deficiencies of the existing varieties and landraces. Variety breeding is very expensive and long lasting. It would also be interesting for the authors to mention the reasons they identified for not cultivating pumpkins in the tropics.

Response: Yes, genetic diversity in pumpkin has been sufficiently studied depending on the agro-ecological regions.  However, in terms of tropical condition the existing varieties cannot cope with the goal of extending this species. This is due to the climate change and emerging environmental challenges such as unpredictable rainfall patterns, excessive temperature, pests and diseases, price fluctuations, detriment lack of incentives like pesticides, thieves, post-harvest losses and difficult to transport. Moreover, general growers and peoples are still unknown on this crop nutritional values and economical importance.

  1. Chapter 3. "Origin and distribution of pumpkin" mentions "A couple of countries have wideranging collections of Cucurbita germplasm [21]." Even if the figure is very explicit, I suggest that at least the first 3-4 countries be listed in the text.

Response: Part of countries holding the collections of Cucurbit germplasm were listed in section 3 as per your suggestion.

  1. The "Introduction" states "Pumpkin cultivation is still ignored in most tropical states." Chapter 4 “World scenario of pumpkin production and harvested area” states: “Pumpkin plants are best cultivated in good, well-drained soil during the warm season. The pumpkin fruits are very susceptible to heat and highly unpreserved in warm conditions [36]. ”the paper would need to outline the area where the pumpkin could spread to tropical states.” The idea must also continue with recommendations on how to spread this crop. To whose detriment do the authors recommend that pumpkins be grown in the tropics? Such recommendations, scientifically documented from all points of view (economic, ecological, nutritional, etc.) could convince the leaders of tropical states to introduce pumpkin in their future agricultural programs.

Response:  All the mentioned comments is responded in the main manuscript and highlighted with red color as: According to FAOSTSA [42] due low production of pumpkin, there is an opportunity to spreading this crop cultivation and utilization in the tropical region such as Mali, Thailand, Malawi, Malaysia, Djibouti, Barbados, Antigua and Barbuda and Dominica.  

Moreover, the recommendation is added in the main manuscript under section 10 as per your suggestion.

Reviewer 2 Report

Major comments:

1)Pumpkin as a crop of high nutritional value- but it should be mentioned that the main carotenoids of pumpkin are not only beta-carotene, but also lutein and zeaxanthin. The authors speak only about beta-carotene and the possibility to treat night blindness caused by vitamin A (beta-carotene) deficiency. In fact lutein and zeaxanthin are not less important because they  are the main carotenoids of a yellow spot in eye retina, thus providing color vision. In this respect, pumpkin may be considered as an excellent product to combat macular dystrophy. Indeed, if you read attentively ref [10] you will see that the amount of beta-carotene reached 6 mg/100g while total carotenoids level was equal to 35 mg/100 g. Furthermore, add the data of pumpkin carotenoids importance for eye health to Figure 5

2) Taking into account high medicinal properties of C.ficifolia it seems interesting to know prospects of its breeding

3) Are there any data on the value of genotypic coefficient of variation for carotenoids content in pumpkin?

                Minor comments:

  • Section 5 ‘The pumpkin pulp, peel and  seeds  are the richest sources of phytochemicals such as    phenol and flavonoid content and  have antioxidant activity’- change ‘phenol’ to ‘polyphenols’- otherwise the meaning will be lost

-‘for this  Pumpkin  consider as  a valuable  vegetable with medicinal and functional culinary properties’- change ‘Pumpkin’ to ‘pumpkin’

-‘[48], [22]’ change to ‘[22,48]’ and below: [51],[10]’ change to ‘[10,51]’+ the same in Section 6 :’[53], [54], [35]’ change to ‘[35,53,54]’ +section 8: ‘[67], [68]’ change to [67,68]+ the same in Tables 1, 2 and 3

-Pumpkin cultivation  primarily  depends  for its edible oil’ change ‘for’ to ‘on’

2)  ref [10]- add volume number

3)Section 8.1 ‘Furthermore, quite insufficient based on the superiority and quality features of the fruit.’- there is no predicate

-‘a breeder is  required to produce’ change to ‘a breeder is  to produce’

4)’ Therefore,  to determine genetic diversity, heritability, genetic development, and the variability (genotypic as well as phenotypic coefficients of variation) of the desired attributes more crucial.’- no predicate

Above Table 3 change ‘utilisation’ to ‘Utilization’

5) Table 3 column ‘crops’- as in all cases ‘pumpkin’ is written may it will be better to substitute this word with ‘C.maxima’, or ‘C.moschata’.. or just delete this column

6) Table 5- change (Sharma, 2006) to [126]

-all Tables- Is column 1 necessary? I would recommend to delete it as it brings no information

7) check Figure numbers: â„–10-11-are absent. The same is with Tables: Table 4 is absent

8) check reference list: year of publication should be written in bold letters (see references 14,21,26,32,56,71,74,79,109,110,115,123-126,128,130,159,160)

9) add volume number to references 12 and 20

10)ref 132: Haq M.A. should be changed to ‘Haq M.M.

11) ref.142 add ‘John Willey and Sons Inc., New York’ and delete ‘allard’

12)ref 120 ‘Cucurbita moschata’ and ‘Cucurbira pepo’- use Italics

Author Response

Response to Reviewer 1 Ref: horticulturae-1353785 MS Title: Pumpkin (Cucurbita spp.): A crop to mitigate food and nutritional challenges, its genetic diversity and improvement revealed by hybridization breeding through diallel mating design – a review Thank you very much for your valuable criticism and suggestion regarding this manuscript. As per reviewer comments and suggestion the manuscript has been revised properly through track change in the main text of the manuscript as well as the point-to-point response of reviewer 1 is given bellow. Major comments Point 1: Pumpkin as a crop of high nutritional value- but it should be mentioned that the main carotenoids of pumpkin are not only beta-carotene, but also lutein and zeaxanthin. The authors speak only about beta-carotene and the possibility to treat night blindness caused by vitamin A (beta-carotene) deficiency. In fact, lutein and zeaxanthin are not less important because they are the main carotenoids of a yellow spot in eye retina, thus providing color vision. In this respect, pumpkin may be considered as an excellent product to combat macular dystrophy. Indeed, if you read attentively ref [10] you will see that the amount of beta-carotene reached 6 mg/100g while total carotenoids level was equal to 35 mg/100 g. Furthermore, add the data of pumpkin carotenoids importance for eye health to Figure 5 Response: pumpkin total carotenoids Importance for eye health is “anti-macular dystrophy” is added to the Figure 5. According to Yadav et al,. (2010), Pumpkin pulp contain antioxidant such as carotenoid, lutein and zeaxanthin and this information is added I section 5 (Figure) and 6. Point 2: Taking into account high medicinal properties of C.ficifolia it seems interesting to know prospects of its breeding Response: The medicinal properties of C.ficifolia is “Anti-hyperglycaemic” is added to the Figure 5. The bellow statement reported by Xia and Wang, (2006) regarding medicinal properties of C.ficifolia is added in main text (section 6). The modification is highlighted by red colour text) “Cucurbita ficifolia is an important species of the Cucurbitaceae family and it is the natural source of insulin mediator namely D-chiro-inositol (D-CI) that acts as an Anti-hyperglycaemic to reduce the blood glucose concentration in Type-2 diabetic patients [54]. Point 3: Are there any data on the value of genotypic coefficient of variation for carotenoids content in pumpkin?
Response: In Table 2 (Row number-4 and 6, column-3; reference number 86 and 90) the traits carotenoids content of pumpkin is added because, this trait showed high value of genotypic coefficient of variation (GCV). Minor comments Point 1: Section 5 ‘The pumpkin pulp, peel and seeds are the richest sources of phytochemicals such as phenol and flavonoid content and have antioxidant activity’- change ‘phenol’ to ‘polyphenols’- otherwise the meaning will be lost Response: the word ‘phenol’ changed to ‘polyphenols’ Point 2: - ‘for this Pumpkin consider as a valuable vegetable with medicinal and functional culinary properties’- change ‘Pumpkin’ to ‘pumpkin’ Response: the word ‘Pumpkin’ changed to ‘pumpkin’ Point 3: -‘[48], [22]’ change to ‘[22,48]’ and below: [51],[10]’ change to ‘[10,51]’+ the same in Section 6 :’[53], [54], [35]’ change to ‘[35,53,54]’ +section 8: ‘[67], [68]’ change to [67,68]+ the same in Tables 1, 2 and 3 Response: all change is done as per your suggestion. Point 4: -Pumpkin cultivation primarily depends for its edible oil’ change ‘for’ to ‘on’ Response: correction done Point 5: ref [10]- add volume number Response: Volume number 45(6) is added Point 6: Section 8.1 ‘Furthermore, quite insufficient based on the superiority and quality features of the fruit.’- there is no predicate Response: The sentence is deleted Point 7: ‘a breeder is required to produce’ change to ‘a breeder is to produce’ Response: The word “required” deleted Point 8:’ Therefore, to determine genetic diversity, heritability, genetic development, and the variability (genotypic as well as phenotypic coefficients of variation) of the desired attributes more crucial.’- no predicate Response: The sentence is deleted Point 9: Above Table 3 change ‘utilisation’ to ‘Utilization’ Response: The ‘utilisation’ changed to ‘Utilization’
Point 10: Table 3 column ‘crops’- as in all cases ‘pumpkin’ is written may it will be better to substitute this word with ‘C.maxima’, or ‘C.moschata’.. or just delete this column Response: the column is deleted Point 11: Table 5- change (Sharma, 2006) to [126] Response: the reference (Sharma, 2006) change to [126] Point 12: -all Tables- Is column 1 necessary? I would recommend to delete it as it brings no information Response: deleted the column 1 from all the tables Point 13: check Figure numbers: â„–10-11-are absent. The same is with Tables: Table 4 is absent Response: Point 14: check reference list: year of publication should be written in bold letters (see references 14,21,26,32,56,71,74,79,109,110,115,123-126,128,130,159,160) Response: year of publication of reference is bolded Point 15: add volume number to references 12 and 20 Response: Correction done for 12 (10.1080/10408398.2021.1896472), for 20: 27 (5) Point 16: ref 132: Haq M.A. should be changed to ‘Haq M.M. Response: changed by Haq M.M Point 17: ref.142 add ‘John Willey and Sons Inc., New York’ and delete ‘allard’ Response: deleted ‘allard’ Point 18: ref 120 ‘Cucurbita moschata’ and ‘Cucurbira pepo’- use Italics Response: changed by italic

Reviewer 3 Report

The effort made by the authors is very valuable, as nowadays it is impossible to follow most of the current literature on a topic of interest. This paper is a comprehensive review focused on the potential uses of pumpkins. The manuscript fits within the scope of the journal. The manuscript is interesting and the idea is nice. The title is clear and it is adequate to the content of the article. The author’s work on discussing achieved results is appreciated. Minor revisions are necessary to improve the clarity of the presentation.

I have some minor recommendations for authors:

Please include some information about the work method: How do you search literature data? How was the period? Which sources?

I'm not sure if so more unpublished pictures it is ok.

Moderate English changes required

Check the bibliography and all text to be written according to the requirements of the journal.

Author Response

Response to Reviewer 2 Ref: horticulturae-1353785 MS Title: Pumpkin (Cucurbita spp.): A crop to mitigate food and nutritional challenges, its genetic diversity and improvement revealed by hybridization breeding through diallel mating design – a review Thank you very much for your valuable criticism and suggestion regarding this manuscript. As per reviewer comments and suggestion the manuscript has been revised properly through track change in the main text of the manuscript as well as the point-to-point response of reviewer 2 is given bellow. Point 1: The effort made by the authors is very valuable, as nowadays it is impossible to follow most of the current literature on a topic of interest. This paper is a comprehensive review focused on the potential uses of pumpkins. The manuscript fits within the scope of the journal. The manuscript is interesting and the idea is nice. The title is clear and it is adequate to the content of the article. The author’s work on discussing achieved results is appreciated. Minor revisions are necessary to improve the clarity of the presentation. Response; Thanks for your critical assessment. I have some minor recommendations for authors: Point 2: Please include some information about the work method: How do you search literature data? How was the period? Which sources? Response: Information taken from published paper, previous research work, published books, Library, Universiti Putra, Malaysia. Authors have tried to collect information and statistical data science 10 years, however, there are some theoretical information was provided as per published year (above ten years). Source of information: Google scholar, several rebound journal. Point 3: I'm not sure if so, more unpublished pictures it is ok. Response: I think it is relevant to my manuscript, which will enrich the overall content. The all-unpublished pictures included in the manuscript has been taken from Author’s research field. Point 4: Moderate English changes required Response: The Manuscript is edited by native English expert. The change is highlighted by red colour in the body of the text. Point 5: Check the bibliography and all text to be written according to the requirements of the journal. Response: The bibliography and all text written in Manuscript is checked properly.
